# Hepatitis C Reinfection in People Who Inject Drugs in Resource-Limited Countries: A Systematic Review and Analysis

**DOI:** 10.3390/ijerph17144951

**Published:** 2020-07-09

**Authors:** Abbe Muller, David Vlahov, Matthew J. Akiyama, Ann Kurth

**Affiliations:** 1School of Nursing, Yale University, 400 West Campus Drive, Orange, CT 06477, USA; david.vlahov@yale.edu (D.V.); ann.kurth@yale.edu (A.K.); 2Department of Epidemiology-Microbial Diseases, Yale School of Public Health, New Haven, CT 06510, USA; 3Department of Medicine, Divisions of General Internal Medicine & Infectious Diseases, Albert Einstein College of Medicine, Montefiore Medical Center, Bronx, NY 10461, USA; makiyama@montefiore.org

**Keywords:** hepatitis C, reinfection, people who inject drugs, upper-middle income country, lower-middle income country, low-income country, direct acting antiviral

## Abstract

Hepatitis C (HCV) is a global pandemic. The World Health Organization has developed a strategic plan for HCV elimination that focuses on low- and middle-income countries (LMICs) and high-risk populations, including people who inject drugs (PWID). While direct-acting antiviral (DAA) therapies are highly effective at eliminating HCV infections and have few side effects, medical professionals and policymakers remain concerned about the risk of reinfection among PWID. This study is a systematic review of research measuring the rate of HCV reinfection among PWID in LMICs and identifies additional areas for further research. A systematic search strategy was used to identify studies documenting HCV reinfection after sustained virologic response in PWID in LMICs. We refined results to include studies where at least 50% of participants had DAA treatment for primary HCV infection. Pooled reinfection rate was calculated across all studies. Seven studies met eligibility criteria. Most studies were conducted in six upper middle-income countries (Mexico, Romania, Russia, Taiwan, Georgi, and Brazil) and one lower middle-income country (Bangladesh) with a total of 7665 participants. No study included information from PWID in low-income countries. Sample sizes ranged from 200 to 3004 individuals, with demographic data missing for most participants. Four studies used deep gene sequencing, and reflex genotyping procedures to differentiate reinfection (infection by a different HCV genotype/subtype) from virologic relapse (infection by the same strain). The follow-up time of people cured from primary chronic HCV infection ranged from 12 weeks to 6.6 years. The pooled reinfection rate of all seven studies was 2.8 (range: 0.02 to 10.5) cases per 100 person-years (PY). In the five studies that differentiated relapse from reinfection, the incidence of reinfection was 1.0 per 100 PY. To date, research on reinfection rates among PWID in LMICs remains limited. Research focused on PWID in low-income countries is particularly needed to inform clinical decision making and evidence-based programs. While rates of reinfection among PWID who complete DAA treatment in upper and lower middle-income countries were similar or lower than rates observed in PWID in high-income countries, the rates were highly variable and factors may influence the accuracy of these measurements. This systematic review identifies several areas for continued research. Policies concerning access to HCV testing and treatment should be comprehensive and not place restrictions on PWID in these settings.

## 1. Introduction

Direct acting antiviral (DAA) regimens have radically changed the treatment of hepatitis C virus (HCV) infection. Introduced in 2014, these highly effective regimens are orally administered and, compared to earlier options, provide a shorter and more easily tolerated course of treatment. DAAs have changed the landscape of HCV treatment by expanding the scope of care to a wider range of patient populations, including people who inject drugs (PWID) [1].

Despite these advances, detectable viral loads following HCV treatment (also known as recurrence) have been an ongoing concern and complicate epidemiological research as the root cause of recurrence can vary. Recurrence is a term that refers to a presence of HCV RNA in the blood without differentiating between reinfection and relapse [2]. Reinfection occurs when a strain of HCV distinct from the primary infecting strain is detected after HCV cure, or sustained virologic response (SVR) [2]. In contrast, relapse occurs when there is a decreased circulation of HCV RNA that remains below the limit of detection in peripheral blood samples during treatment but rebounds after treatment cessation [3]. Spontaneous clearance, or the elimination of virus without the use of medication, occurs around 25% of the time with primary infection and around 40% in reinfection [4]. Differentiating between reinfection and relapse can only be definitively done through advanced sequencing methods that analyze the HCV strain and compare results to the primary infection.

Emboldened by the entry of DAA regimens, the World Health Organization (WHO) and member states initiated in 2016 the Global Health Sector Strategy (GHSS) for Viral Hepatitis to eliminate HCV by 2030, with a focus on low and middle-income countries (LMICs) [5]. The GHSS’s First Strategic Direction aims to increase information for focused action (identifying hepatitis epidemics and the target populations to treat), and the Second and Fourth Strategic Directions focus on defining high-impact interventions and increasing these interventions’ cost-effectiveness to promote sustainability, respectively [5]. Despite the shared global aim of elimination, funding and aid given to these LMICs have declined, resulting in decreased funding for new and existing health programs.

Clinicians and policy makers have expressed concerns that PWID are likely to have low treatment adherence and high rates of reinfection [1,6]. In one study, only a small proportion (15%) of surveyed clinicians reported they would initiate DAA treatment in PWID [1]. Conversely, authors of another qualitative study found that the combination of stigma-free medical care with social services reinforces the provider–patient relationship and enhances the positive consequences of treatment, including adherence to substance use disorder treatment and housing/employment stability [7]. This hesitancy among providers and policymakers to provide PWID with HCV treatment is problematic as it not only exacerbates existing stigma and health disparities, but it also fails to promote effective public health strategies. PWID represent the highest proportion of incident HCV cases and failing to treat PWID can maintain, or even expand, a local epidemic. Therefore, dedicated research in resource-limited settings, including on rates of reinfection, is necessary to better understand the HCV epidemic among PWID in LMICs.

One of the main challenges in defining the rate of HCV reinfection is the low availability of clinical data in resource-limited settings. Epidemiological research primarily uses three methods to determine the rate of recurrence. These methods, which differ in accuracy and cost, include deep gene sequencing, also known as next-generation sequencing, simple genotyping, and measuring quantitative RNA load. The “gold standard” (the method with the least uncertainty) involves the deep sequencing of the HCV genome before and after SVR to distinguish reinfection by a separate viral strain from the primary infection; however, the high cost of deep sequencing may preclude its use in resource-limited settings [8,9].

Simple genotyping identifies segments of the highly variant regions (~35% variability) of the viral RNA to classify the tested strain as one of seven major types and 67 subtypes [10,11]. Simple genotyping of primary infection and reinfection is limited though as it can only distinguish viral strains if the individual is not reinfected with the same strain. In certain geographical areas, the number of different strains is small. For example, among the Kenyan PWID population, two strains (1a and 4a), are dominant [12]. A limited number of circulating strains can reduce the accuracy of both the measurement of reinfection and the sensitivity for classification of reinfection vs. relapse [13]. Without the ability to distinguish relapse from reinfection, the quality and outcomes from HCV treatment may be difficult to measure; if this issue is systemic, meeting WHO GHSS Strategic Directions may be challenging [14]. Even with the advent of pangenotypic regimens, relapse is often treated with a different protocol (drug and length of time) from that used to treat reinfection. Despite these limitations, simple genotyping is less costly than deep gene sequencing.

The third method involves measuring the HCV RNA viral load or calculating the copies of virus per milliliter of blood. While accurate and reliable, this method does not differentiate reinfection from relapse [8,9]. Despite its significant limitations, HCV viral load testing is the least expensive method of determining recurrence and is the most widely available measurement globally. This highlights the need for inexpensive laboratory and field-testing methods to cost-effectively differentiate relapse from reinfection to guide clinical care.

In conducting a review of HCV reinfection among PWID, caution is required when defining the outcome variable (reinfection) and analyzing which instruments or assays are used to identify the outcome variable. Simmons and colleagues (2016) reported a meta-analysis of HCV reinfection studies among PWID in high-income countries, where they used a broad definition of recurrence (“confirmed HCV RNA detectability post-SVR”) but made the relapse vs. reinfection classification dependent on terminology used in the original studies [15]. Of the 59 studies included in the meta-analysis, Simmons et al. reported that 23 used genotyping (including 9 of 13 studies in PWID currently injecting), 5 studies used author judgement of individual patient risk, and 31 did not specifically classify or distinguish relapse from reinfection [15]. Authors of a recent meta-analysis of HCV reinfection in PWID included 36 studies and 6311 person-years (PY) of follow up and found an overall reinfection rate of 5.9 per 100 PY (recent drug use 6.2/100 PY, opioid substitution therapy (OST) 3.8/100 PY) [16]. Only two of the 36 studies (one of which was a multi-country trial) included upper middle-income countries; the remainder only drew data from high-income countries.

To date, research on high-risk populations such as PWID has focused mostly on high-income countries [17,18]. The WHO has identified a present-day lack of research, interventions, and policies for PWID at risk of or living with HCV, particularly among those in African and Asian LMICs [19]. To better understand the scope of available research regarding HCV reinfection among PWID in resource-limited settings, in this study we systematically reviewed published data on HCV reinfection in LMICs.

## 2. Methods 

### 2.1. Search Strategy

The search strategy included index identification—PubMed, Web of Science, OVIDEmbase, OVID Global Health, CINAHL and Africa Wide Information. We conducted a grey literature search in Google Scholar and clinicaltrials.gov. A sample search for the Medline database is included as Appendix A. We used a search strategy of controlled vocabulary terms and synonymous free text to capture the concepts of hepatitis C, reinfection, recurrence, and drug use by people who inject drugs. We excluded articles published before 2013 because all oral DAA regimens were not available in this period. We scanned titles and abstracts to eliminate studies conducted in high-income countries and those not using DAA treatment regimens. We conducted an additional review of the LILAC database-a database for Latin American and Caribbean health science literature, due to a lack of published studies in Latin and Central America and the Caribbean. Our Ovid MEDLINE search is shown in Figure 1. The conduct and reporting of this study were guided by PRISMA and the full strategy is available from the lead author.

### 2.2. Inclusion and Exclusion Criteria

For this review, we set the date for the searches as of February 2020. Only studies published in 2013 or later were included, which reflects the period when DAAs were introduced. Additionally, we only included studies conducted in LMICs (or multisite studies that contained a substantial number of participants from LMICs), published in English, with a sample size greater than 50, and an attrition rate below 20% [27]. We excluded articles focusing on incarcerated and transplant populations, because the prison environment and transplant protocols yields data that would not be generalizable to the majority of the PWID population. We accepted studies in which participants completed treatment with any interferon-free DAA regimen. Studies whose participants underwent a combination of interferon-free and interferon-containing regimens were included so long as interferon-free treatment regimens comprised over half of the cases. Participants in the studies included persons who injected drugs and who achieved SVR (undetectable HCV RNA) at least 12 weeks following treatment completion. Table 1 shows sampling, the rate of reinfection (over 5 cases per 100PY constituted “high”) and recruitment methods in addition to assays used and testing interval. In cases where methodological information was missing, we sent inquiries to authors to provide supplemental data. Such inquiries resulted in one author providing supplementary documents. Reinfection rates were abstracted directly or calculated (the number of reinfections divided by total PY) from data included in the reports. The results obtained from our search were initially screened by abstract and title and are represented by the PRISMA flowchart in Figure 1. The full-text review excluded articles that did not meet inclusion criteria (high-income country, interferon containing treatment, no PWID, etc.) or did not include recurrence data. In the case of multiple publications of a single study, we included the study with most up-to-date data. Attrition rates were calculated (dropouts/total) from supplemental information if not provided in the manuscript. Systematic reviews and meta-analyses were included if they contained unpublished data or unpublished studies. 

## 3. Results

Seven studies met inclusion criteria. Table 1 provides the design, key characteristics, and HCV reinfection rates for each included study. The studies had three different designs: cohort study, clinical trial, and integrated analysis (systematic review, meta-analysis). Five of the seven studies were conducted across multiple sites, including the following lower middle-, and upper middle-income countries: Mexico, Romania, Russia, Taiwan, Georgia, Bangladesh, and Brazil. These multi-site studies also used data from partnering high-income countries, including Canada, Australia and the United States. We observed an inverse association between HCV infection rates and country income level (World Bank category and median income). While the available data are insufficient to comment further, this finding merits further investigation. No studies were conducted in low-income countries. Study population sizes ranged from 200 to 3004 participants. However, the largest study of 3004 individuals did not report demographic information. Among studies reporting demographic information, participants were mostly male (range: 57% to 100%). The length of follow-up in these studies ranged from 2 to 62 months.

The overall crude reinfection rate for the pooled data was 2.8 cases per 100 PY (95% CI 0.8 to 6.3). Table 1 shows that only the Huang (2019) study had [21] a reinfection rate greater than 5 per 100 PY, and this study only analyzed recurrence (no differentiation between relapse and reinfection); five studies had rates under 2 per 100 PY. Only the Latham (2019) study compared participants using and not using OST services, and found a lower rate of reinfection (0.6 cases per 100PY for OST recipients and 1.9 cases per 100PY for non-OST recipients). The prevalence of HCV in the study populations (when reported or available) ranged from 33.7% to 56.8%. No report included background information on recruitment methods (e.g., flyer, bus ad, outreach in clinics) or sampling methodologies (e.g., consecutive patients); inquiries for supplemental information yielded one result. Testing intervals ranged from varied or not delineated in three studies [21,22,25] to a single testing time point in three studies [20,23,26]; only two studies delineated multiple testing time points over a significant follow-up period (2–6 years).

## 4. Discussion

While the results from the studies in our systematic review were heterogeneous, we found that reinfection rates measured in studies of PWID in upper and lower middle-income countries were comparable or lower than the reinfection rates of PWID in high-income countries. Hajarizadeh and colleagues reported a meta-analysis on PWID who completed DAA treatment; they identified studies conducted in Demark, Spain and the USA with high reinfection rates ranging from 8.0 to 18.3 cases per 100 PY and studies with low reinfection rates from 1.1 to 1.7 cases per 100 PY in Canada, the USA and Germany [16]. Notably, the USA study results are in both the low and high reinfection groups; the results come from different studies in different geographic regions which may reflect local conditions and variations between populations. Though we observed this inverse relationship between reinfection rates and country income level, further research is needed in LMIC settings. The variance among these studies, along with the absence of studies measuring reinfection rates among PWID in low-income countries, underscore the need for more large-scale observational studies dedicated to PWID in LMICs. Consistent with findings in high-income countries, one study found a reduction in reinfection in OST recipients compared to non-OST recipients, indicating that the beneficial impact of harm reduction services is consistent regardless of economic status.

Based on our systematic review, we also suggest that certain factors may influence the measured reinfection rates seen among PWID in upper and lower middle-income countries. Past epidemiological studies have shown that higher background HCV prevalence is associated with a higher risk of reinfection rates among people actively engaged in high-risk injection behavior (e.g., sharing needles or equipment) [28]. Modeling studies have applied this understanding and generally agree that “treat all” mandates are highly cost-effective if the background HCV prevalence in the source population is below 50% [28,29]. Among studies in our review, however, background prevalence did not appear to have a significant impact on reinfection rates. In fact, as seen in Table 1, the study with the highest reinfection rate had the lowest background HCV prevalence.

Our review of studies suggests that the testing interval for recurrent HCV viremia (e.g., every 24 weeks versus 12 weeks) may play a role in reinfection rate. Due to viral clearance of the reinfection, longer testing intervals between assessments for recurrent viremia can result in lower measured reinfection rates. Among the reviewed studies, longer testing intervals did appear to be associated with lower reinfection rates supporting this finding. Similarly, the length of follow-up may play a role in reinfection rate. While the role of longer follow-up needs to be clarified, lower reinfection rates may be observed because those who are at the highest risk of reinfection due to high-risk behavior or being in high-risk networks tend to get reinfected early [30,31]. Accumulating more person-years of follow-up with fewer additional reinfections over time and potential losses to follow-up of the highest risk individuals may also lead to a lower reinfection rate.

Notably different among the studies was the use of genotype or deep sequencing in distinguishing reinfection from relapse. Genotyping, and particularly deep sequencing, in epidemiological research allows for a closer examination of the factors associated with HCV transmission, such as sharing injecting equipment, the concurrent use of other substances, and unprotected sex. Such molecular epidemiologic tools can enhance our understanding of factors associated with reinfection. Our review did not have enough studies using genotyping to draw large-scale conclusions about factors that play a role in reinfection vs. relapse. Broader use of these tools is recommended.

Our study highlights the current lack of data about HCV reinfection rates in low-income countries and the limited data available about HCV reinfection rates in middle-income countries. This deficiency undermines policy makers’ efforts to develop clinically appropriate treatment guidelines and implement evidence-based programs. The lack of data also weakens global efforts to address socioeconomic health disparities. Given the limited data on HCV reinfection rates in PWID in resource-limited settings, policy makers should commit to using comprehensive programs that implement a broad range of public health strategies supported by most current modeling studies, including widespread testing, faster linkage to care, treatment for all, and increased access to harm reduction interventions [32,33,34,35,36]. These interventions should be combined with efforts to specifically engage PWID in resource-limited countries and further our understanding of the HCV epidemic.

This study has some limitations. First, there were a limited number of studies outlining reinfection rates in middle-income countries and no studies in low-income countries. Second, heterogeneous study designs, the variability of follow-up time, and different methodologies to measure reinfection presented significant challenges to comparing or allowing a confident summary of HCV reinfection rates in these settings. Finally, high-risk characteristics are not always consistent across regions, cultures and economic classes. Despite these limitations, our systematic review demonstrates that HCV reinfection rates among PWID were low overall. These results are equivalent to studies conducted in the general population (not PWID) and are comparable or even lower than those of studies in high-income countries [37,38,39,40,41,42].

## 5. Conclusions

The results of this systematic review highlight the heterogeneity in study design, reinfection measurement, and testing intervals. Overall, reinfection rates ranged from 0.024 cases per 100 PY to 10.5 cases per 100 PY, with an overall pooled crude rate of 2.8 cases per 100 PY. This is the first review to concentrate on PWID in LMICs, providing initial evidence that concern for HCV reinfection should not be used to justify withholding DAA treatment outside high-income counties.

## Figures and Tables

**Figure 1 ijerph-17-04951-f001:**
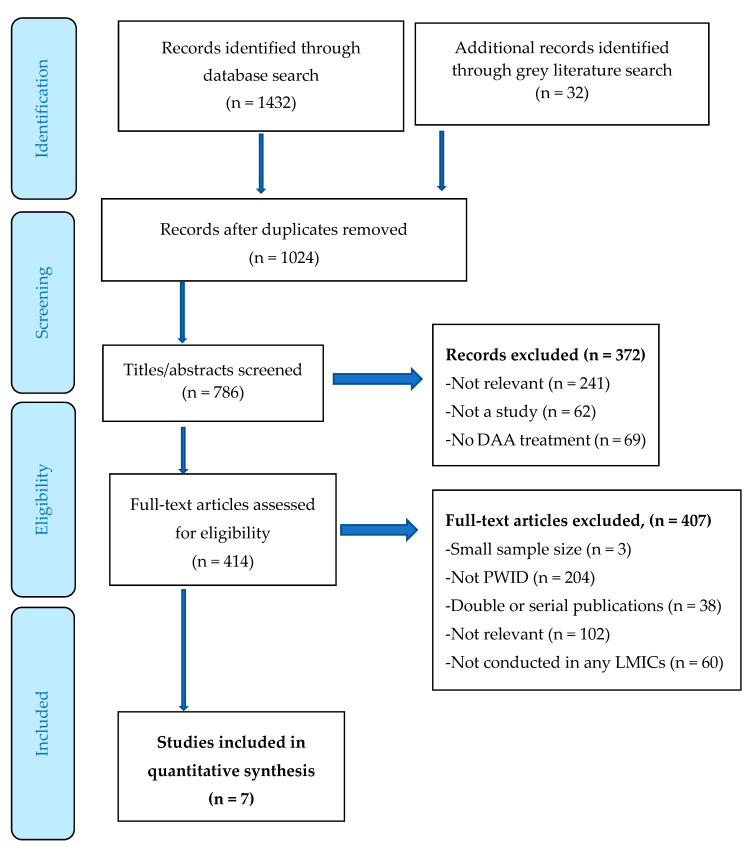
Prisma flowchart.

**Table 1 ijerph-17-04951-t001:** Summary of studies on HCV reinfection in low- and middle-income countries.

	Author, Year	Study Design	Setting (LMICs in Multi-Site Studies, SES and Median Income of Relevant Countries)	Study Population (N) % Male Initial HCV Prevalence	Testing Interval	Reinfection Rate (per 100 PY) (Available Rates among Relevant Special Populations Noted)	Measurement of Reinfection	Follow-Up Time (Participant Range if Given)	Attrition Rate (Loss to Follow-Up)
1	Foster et al., 2019 [20]	Integrated analysis of clinical trials	Multi-country Upper middle-income (Mexico = $9673 Romania = $12,306 Russia = $11,288)	N = 1819 Male 57.0% Prevalence = 50.0% *	Followed for 24 weeks, reinfection detected at week 12	0.6	Deep gene sequencing	24 weeks	0.01
2	Huang et al., 2019 [21]	Cohort study	Taiwan = $14,273 Upper middle-income	N = 219 Male 100.0% Prevalence = 33.7%	Varied: either 12 months (majority), following abnormal labs (minority)	10.5 14.1 (for DAA treatment recipients)	No differentiation between reinfection and relapse	2.1–6.6 years	0.04
3	Latham et al., 2019 [22]	Systematic review and meta-analysis	Multi-country Upper middle-income (Georgia = $4722)	N = 827 Sex not delineated Prevalence not delineated	Varied among studies	1.9 (for recent PWID) 0.6 (for OST recipients)	Genotyping, deep gene sequencing or none	24 weeks–3 years	0.02
4	Rahman et al., 2019 [23]	Prospective cohort	Bangladesh = $1698 Lower middle-income	N = 200 Sex not delineated Prevalence = 42%	Once 12 weeks after SVR	4.2	Genotyping	12 weeks	0.05
5	Reddy et al., 2018 [24]	Cohort study	Multi-Country Upper middle-income (Brazil = $9001, Argentina = $11,683, Mexico = $9673, Taiwan = $14,273)	N = 1503 Male 60% Prevalence = 56.8% *	Day 1, week 24, 48, 96 and 144	0.02	Reflex genotyping	144 weeks	0.01
6	Rockstroh et al., 2017 [25]	Clinical trial	Multi-Country Upper middle-income (Russia = $11,288)	N = 228 Male 80%Prevalence = 50% *	Not delineated	1.9	Deep gene sequencing	12 weeks	0.02
7	Sarrazin et al., 2017 [26]	Cohort Study	Multi-Country Upper middle-income (Russia = $11,288)	N = 3004 Sex not delineated Prevalence = 50% *	Once at 24 weeks	0.5	Deep gene sequencing	24 weeks	Not delineated

***** background prevalence average among all sites in multi-site studies.

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
