# Peer review of "Hepatitis C Reinfection in People Who Inject Drugs in Resource-Limited Countries: A Systematic Review and Analysis"

_ijerph, 2020, doi:10.3390/ijerph17144951_

Round 1

Reviewer 1 Report

The authors conduct a literature search in order to assess the policies recommended by the WHO for the control of HCV, with focus on PIWD in low and middle income countries. The authors provide a good background of the factors involved and the caveats of these kinds of epidemiological studies. 

Major concerns:

The rate of re-infection among PWID in middle and low income countries is relevant when is compared to either the rate of new infection among PIWD or compared to the rate of re-infection among PIWD in high income countries. These numbers need to be provided with the relevant studies and the numbers of patients in these studies. 

In general the authors have an extensive discussion but do not show the results that lead to this discussion.

The authors need to have more complete tables.  The number of patients in each study and the actual recurrence/re-infection rates should also be provided. There should also be a more defined interval between SVR and reinfection. 

Does the high rate of re-infection correlate with either median income or with the time between SVR and sampling? This in not clear, nor is the information needed to assess this by the reader available. It would also be of interest to show median income of either the study patients or the relevant country, again so the reader can assess it. 

line 62-63: the authors state the re-infection rate in lower and middle income countries are comparable to high income countries but no data or references are provided. Please include this data along with patient numbers. Please change high/low to actual numbers, or at least state what the boundaries are. 

line 70-71 the authors state the re-infection rate is higher among people actively engaged in high risk behaviour.  Is this also true for the de-novo rate of infection? This is relevant for people interested in vaccination. 

line 75-76 the authors state the background rate did not appear to have significant impact on reinfection rates. Please provide the data in the results section or tables.

Line 78-81 The authors state that the testing interval may be the cause of the high or low re-infection rates again the numbers should be provided.  What was the median/mean time between SVR and re-infection in each of the studies ?

Which of the factors mentioned has the greatest impact on re-infection rate?

There is a section for Conclusions that is not completed. Please complete. 

Author Response

Thank you so much for your comments.  I have provided my responses in bold and highlighted discussion changes in yellow in the manuscript.

The rate of re-infection among PWID in middle and low income countries is relevant when is compared to either the rate of new infection among PIWD or compared to the rate of re-infection among PIWD in high income countries. These numbers need to be provided with the relevant studies and the numbers of patients in these studies. 

In general the authors have an extensive discussion but do not show the results that lead to this discussion.

The authors need to have more complete tables.  The number of patients in each study and the actual recurrence/re-infection rates should also be provided. There should also be a more defined interval between SVR and reinfection. 

Response: Our Table 2 was not included in the manuscript I received, I merged our Table 2. with Table 1 in the main text.  This new Table 1 includes follow-up time, reinfection rate, background prevalence, measurement of reinfection, type of study and setting.  This table helps to bridge the results to the discussion.

Does the high rate of re-infection correlate with either median income or with the time between SVR and sampling? This is not clear, nor is the information needed to assess this by the reader available. It would also be of interest to show median income of either the study patients or the relevant country, again so the reader can assess it. 

Response: Shorter time from SVR to sampling for reinfection has been associated with higher rates of reinfection (see lines 83-88). Demographics on reinfected cases were often omitted from the studies; however, we have added country median income (GDP per capita) to Table 1.

line 62-63: the authors state the re-infection rate in lower and middle income countries are comparable to high income countries but no data or references are provided. Please include this data along with patient numbers. Please change high/low to actual numbers, or at least state what the boundaries are. 

Response: We have added a paragraph in the discussion section summarizing a meta-analysis range of reinfection within PWID in high income countries (see lines 214-218).

line 70-71 the authors state the re-infection rate is higher among people actively engaged in high risk behavior.  Is this also true for the de-novo rate of infection? This is relevant for people interested in vaccination. 

Response: This is consistent with primary HCV infection as well,

line 75-76 the authors state the background rate did not appear to have significant impact on reinfection rates. Please provide the data in the results section or tables.

Response: We have the added information to Table 1 which includes background prevalence of HCV in the fourth column (refer to the Huang study which has the lowest background prevalence and highest reinfection rate).

Line 78-81 The authors state that the testing interval may be the cause of the high or low re-infection rates again the numbers should be provided.  What was the median/mean time between SVR and re-infection in each of the studies?

Response: Follow-up time is included in the added information in Table 1 (in second to last column) and often represents testing interval.  Calculating median time to reinfection is difficult given the little demographic data reported on cases.

Which of the factors mentioned has the greatest impact on re-infection rate?

Response: We appreciate this question. Due to the heterogeneity in design and methods of the studies included in this review, it is difficult to compare which factor has the greatest impact on re-infection rate.

There is a section for Conclusions that is not completed. Please complete. 

Response: I added the conclusion section and changed the citation format to IJERPH template.

Reviewer 2 Report

This paper was well written, interesting to read, and clearly highlighted the need for more research to be done with this vulnerable population.

Table 1- it would be helpful to have a column stating how many subjects were involved. I am unclear of the relevance of the recruitment/sampling method as PWID are a universally challenging group to recruit for studies. This information could be listed within the manuscript  to make room for the number of subjects.

It appears that the discussion and conclusion were grouped together. Separate the conclusion findings and place under the appropriate header.

Author Response

Thank you so much for your comments.  I have provided my responses in bold and highlighted discussion changes in yellow in the manuscript.

This paper was well written, interesting to read, and clearly highlighted the need for more research to be done with this vulnerable population.

Table 1- it would be helpful to have a column stating how many subjects were involved. I am unclear of the relevance of the recruitment/sampling method as PWID are a universally challenging group to recruit for studies. This information could be listed within the manuscript  to make room for the number of subjects.

Response: We appreciate your question. Table 2 was not included in the manuscript, so we merged our Table 2 with Table 1.  This new table includes follow-up time, reinfection rate, background prevalence, measurement of reinfection, type of study and setting.  This table helps to bridge the results to the discussion.

It appears that the discussion and conclusion were grouped together. Separate the conclusion findings and place under the appropriate header.

Response: I added the conclusion section and changed the citation format to IJERPH template.

Round 2

Reviewer 1 Report

Thank-you for adding the additional data from the studies being reviewed. Most of my concerns are minor.

Minor:

It appears that a single study may be an outlier.  How is this study different from the others? (Huang). 

Now that there are data, there appear to be several trends that could be emphasized:

1.The inclusion of OST as a treatment has a positive effect similar to what has been seen in other studies. 

2. Except for the Huang study, the rates of reinfection in your analysis are significantly lower than is high income countries. Conversely (again except for the outlier) there is an inverse correlation between income and infection rate, The studies in higher income countries have lower rates of infection (among the studies listed).

In Table 1. This is a much better table now and allows some assessment of the data, however please be consistent in the formatting.  If data is not included please say not delineated (for example in column 4 mean age is not there, prevalence is sometimes there). In column 7 deep_____? please pu sequencing (I assume). In column 6 there is various information in brackets that is valuable but it is not in the legend.  Please put in legend and assess this for the other studies where it is not shown or if it is not delineated then say it is not delineated. 

Line 216 are the inits of re-infection per 100PY? please include them.

Line 214 the reference is 2019 not 2020, this is careless. 

Lines 211-216. I disagree with your conclusion. Huang study the data re-analyzed here point to an income dependancy and given the difference between the upper middle/lower middle income countries used here  and the high income countries elsewhere, these differences are interesting. 

Lines 215-216 The US in present in both the low and high re-infection groups, what is the reason for this and does this shed any light on your studies? 

Major:

I apologize for not seeing this in the first round.  the Lantham study is a meta-analysis itself. What is the reason for not using the primary studies included in their analysis in your analysis, instead of the meta analysis?

Author Response

Letter to Reviewer #1

Minor:

It appears that a single study may be an outlier.  How is this study different from the others? (Huang). 

Response: Thank you for highlighting the Huang study with the highest rate of recurrence (10.5 cases per 100PY) of any study, however, they did not differentiate relapse from reinfection.  This information was added in the text of the results section (lines 206-208).

Now that there are data, there appear to be several trends that could be emphasized:

1.The inclusion of OST as a treatment has a positive effect similar to what has been seen in other studies. 

Response: This effect is seen in the Latham paper, we added a description of this in the results section (see lines 208-210), and a conclusion sentence in the discussion section (see lines 230-232).

  1. Except for the Huang study, the rates of reinfection in your analysis are significantly lower than is high income countries. Conversely (again except for the outlier) there is an inverse correlation between income and infection rate, The studies in higher income countries have lower rates of infection (among the studies listed).

Response: Response: The reviewer raises an intriguing observation, namely, a negative trend between HCV rates and income when looking across High versus Middle income countries and within the Middle Income Countries (the median incomes) We have added a sentence in the manuscript’s discussion to highlight this observation (see lines 198-200, 224-228).  However, due to the small number of studies and the different methodologies used it is not possible to do a formal evaluation of the correlation between income and infection rate.

In Table 1. This is a much better table now and allows some assessment of the data, however please be consistent in the formatting.  If data is not included please say not delineated (for example in column 4 mean age is not there, prevalence is sometimes there). In column 7 deep_____? please pu sequencing (I assume). In column 6 there is various information in brackets that is valuable but it is not in the legend.  Please put in legend and assess this for the other studies where it is not shown or if it is not delineated then say it is not delineated. 

Response: I have reformatted the column to clarify the information for the readers.  We also added to the legend (in the column in parentheses) to include bracketed information in the sixth column.

Line 216 are the inits of re-infection per 100PY? please include them.

Response: Thank you for your comment we added the units of reinfection (cases per 100PY in lines 223-224).

Line 214 the reference is 2019 not 2020, this is careless. 

Response: We have fixed this error.

Lines 211-216. I disagree with your conclusion. Huang study the data re-analyzed here point to an income dependancy and given the difference between the upper middle/lower middle income countries used here  and the high income countries elsewhere, these differences are interesting. 

Response: We appreciate your point and in response to this and the previous comment we wanted to reiterate the conclusions made earlier of the trend between country income and reinfection (see lines 230-232).

Lines 215-216 The US in present in both the low and high re-infection groups, what is the reason for this and does this shed any light on your studies? 

Response: The review makes an astute observation.  We have added a sentence in the manuscript’s discussion (see lines 224-228) the following sentence to address this observation.

Major:

I apologize for not seeing this in the first round.  the Lantham study is a meta-analysis itself. What is the reason for not using the primary studies included in their analysis in your analysis, instead of the meta analysis?

Response: The Latham paper used unpublished data, this was clarified in the narrative of the methods section (see lines 197-199).

Round 3

Reviewer 1 Report

I do not have any further concerns. thank-you.